# Free Space Optical Communication Networking Technology Based on a Laser Relay Station

**Changchun Ding, Chengming Li, Ziming Wang, Zhen Gao, Zijian Liu, Junfeng Song \* and Min Tao \***

State Key Laboratory of Integrated Optoelectronics, College of Electronic Science and Engineering, Jilin University, Changchun 130012, China
\* Correspondence: songjf@jlu.edu.cn (J.S.); taomin@jlu.edu.cn (M.T.)

**Abstract:** Optical communication modulation technology and networking technology are two important technologies for constructing free-space optical (FSO) communication. In this paper, pulse width modulation (PWM) is used to realize free-space optical communication. The process of signal modulation and demodulation is implemented by means of a field programmable gate array (FPGA). An optical communication relay system is constructed to realize communication networking. The binary data bits in the communication process are converted into pulse signals of different widths, the data demodulation process is realized by sampling with a high-speed analog-to-digital converter (ADC), the data level is determined by counting the proportion of high and low voltages sampled in a pulse period. The relay system analyzes the routing target after receiving the pulse signal from the transmitter, and then sends the data to the target receiver. The experimental results show that the constructed system can achieve point-to-multipoint free-space optical communication. Additionally, using ADC to demodulate the received signal increases the stability of the free-space optical communication system. This system provides the design prototype system of FSO communication networking technology.

**Keywords:** free-space optical (FSO) communication; pulse width modulation (PWM); field programmable gate array (FPGA); relay station; communication networking

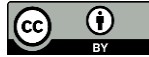

## 1. Introduction

With the development and progress of science and technology, space laser communication technology has gradually expanded from satellites to the ground. At present, laser communication technology has been studied in the fields of lidar, the military, and underwater communication [1–3]. Space laser communication facilitates the exchange of information between two devices without any connection relationship in free-space. During laser communication, the laser transmitting end and the receiving end need to be aligned with high accuracy, and there should be no obstacles blocking the optical path between the two ends; otherwise, data transmission will fail. At the same time, point-to-point laser communication cannot complete point-to-multipoint reception. This greatly increases the difficulty of using laser communication technology in cities and in vehicle-mounted lidars. Therefore, laser communication technology is not widely used in life at present [4–6]. Compared with traditional wireless communication, laser communication does not require complex antennas and frequency applications. At the same time, it has the advantages of large transmission capacity, long transmission distance, good confidentiality, and strong anti-interference ability. Therefore, laser communication technology has broad application prospects in the field of wireless communication [7–9]. Realizing data and information transfer of free-space optical communication and information transmission of one-to-many and many-to-many communication devices is a necessary condition for the application of free-space optical communication technology. Therefore, the

research on free-space optical communication networking technology has important research significance for the wider use of optical communication [10–13].

In the application of a free-space optical communication system, the atmospheric turbulence effect is an important problem to consider [14], due to the changes in wind speed, the unevenness of the sun's radiation energy on the ground and other factors which cause local atmospheric density changes and produce vortexes in the atmosphere. Various vortex superpositions produce atmospheric turbulence, which greatly reduces the communication distance of free-space optical communication and damages the free-space optical communication link between the satellite and the ground, resulting in the attenuation of laser signals. Increasing the gain of the signal at this point does not necessarily improve the quality of the laser beam [15]. However, adaptive optics can effectively improve the effects of atmospheric turbulence or scattering and increase the speed of communication systems [16,17]. Adaptive optics technology is a correction module that adds a conjugate with a distortion phase in front of the communication terminal, thereby suppressing the atmospheric turbulence effect, which is of great significance for the realization of high-rate laser communication such as satellite–ground communication, air–ground communication, and underwater optical communication [18,19].

Free-space optical communication technology needs to load the information to be transmitted onto the laser light wave and send the data in the form of a light wave, which involves converting the data into optical signals (signal modulation) [20,21]. At present, the modulation methods of space laser communication mainly include switch keying (OOK) modulation, pulse interval modulation (PIM), single pulse position modulation (LPPM), differential pulse position modulation (DPPM), pulse amplitude modulation (PAM), etc. [22–25]. Among these modulation modes, the OOK mode has the simplest modulation principle, but its average transmission power is large. The number of timeslots included in each symbol of PIM mode is not fixed, and it has a high channel capacity, but its intersymbol interference is large, and its bit error rate is high. The LPPM mode has a fast transmission rate and low requirements for energy, but it requires high peak power. The DPPM mode has a high-power utilization ratio and frequency band utilization ratio, and a low channel occupation ratio, but its modulation and demodulation are difficult to achieve due to its great difficulty. The PAM mode requires a large transmission power, a complex modulation and demodulation circuit, and a high bit error rate. At the same time, in the above-mentioned modulation methods, it is necessary to ensure that the clocks of the transmitting system and the receiving system are strictly aligned, or data transmission may be misaligned, which may lead to information exchange failure between the two information systems [26–31].

In order to realize point-to-multipoint free-space optical communication on the ground and improve the quality of information transmission, based on the full analysis of the advantages and disadvantages of the above modulation methods, this paper proposes a realization method of free-space optical communication networking technology based on pulse width modulation (PWM). The binary data transmitted by each bit is characterized by sending a pulse signal with a fixed period and an unfixed pulse width into the space. The pulse signal with a narrow pulse width represents datum 1, and the pulse signal with a wide pulse width represents datum 0. At the same time, the receiver samples the signal received by the detector at a fixed frequency through high-speed ADC. In each part of the data modulation period, the width of the modulation pulse is determined by the number of times the ADC samples to a high level. According to the different data frame codes demodulated by the relay system, the data are sent to targets in different directions to achieve point-to-multipoint free-space optical communication. The modulation and demodulation processes are implemented by FPGA [32–36].

## 2. Theory and System Design

A.  Theory of optical communication based on pulse width modulation.

When modulating and demodulating data, one needs to consider the symbol length of the data. If binary data are represented by M bits, this can represent 2M different types of data. If the M is larger, the probability of errors in data modulation and demodulation is greater, and the modulation and demodulation circuits are more complex. In order to build a free-space optical communication system more easily, we choose M = 1 to realize space optical communication. When M = 1, the symbol structure diagram of several free optical communication modulation methods is shown in Figure 1.

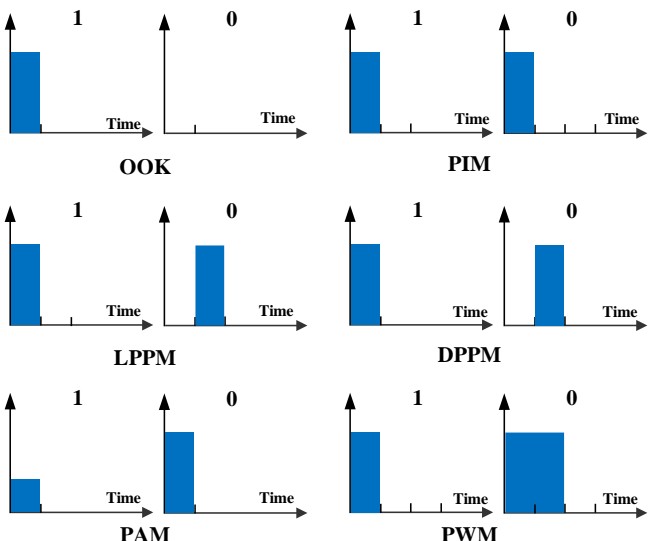

**Figure 1.** Symbol structure of several modulation modes of optical communication when M = 1 [37].

For M = 1, the OOK method only characterizes datum 1 or datum 0 according to whether a pulse signal appears. The number of time slots in PIM mode is not fixed, and it is characterized by datum 1 or datum 0 according to the number of time slots. The LPPM method is characterized by datum 1 or datum 0 according to the different locations where a single pulse appears. The DPPM method is similar to the LPPM method, and is also characterized by datum 1 or datum 0 according to the different positions where a single pulse appears, the difference being that when the pulse signal in the DPMM method appears, it represents the end of a datum—that is, the rest of the time slots are all removed, so the number of time slots in the DPPM method is not fixed. The PAM method is characterized by datum 1 or datum 0 according to the amplitude of the pulse. The PWM method is characterized by datum 1 or datum 0 depending on the pulse width [38–40].

Further analysis of the performance of various modulation methods, assuming the transmission power of the light source, the transmission bandwidth of OOK, and the channel capacity of OOK, is performed according to the principle of optical communication modulation [41–46], and the performance comparison histogram of the average transmission power, transmission bandwidth and channel capacity of various modulation modes when M = 1 is shown in Figure 2. For the data with symbol length M, the average symbol length, average transmit power, transmission bandwidth, channel capacity, and signal synchronization of the above optical communication modulation methods are further summarized according to Figure 2, as shown in Table 1.

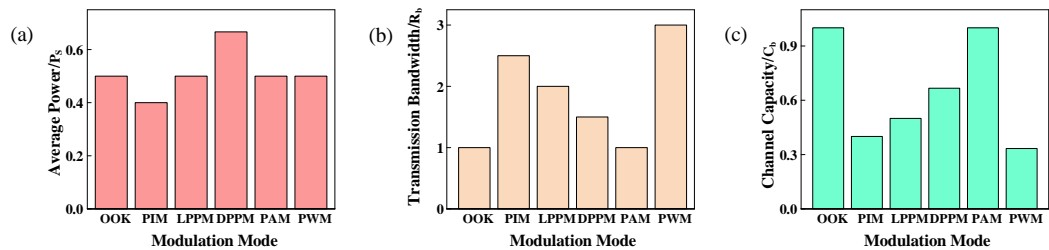

**Figure 2.** Performance histogram of several optical communication modulation methods when M = 1 [47,48]. (**a**) Average power. (**b**) Transmission bandwidth. (**c**) Channel capacity.

**Table 1.** Performance index of several modulation modes in optical communication.

| Modulation Mode | OOK | PIM | LPPM | DPPM | PAM | PWM |
|---|---|---|---|---|---|---|
| Average symbol length | M | $\dfrac{2^M+3}{2}$ | $2^M$ | $\dfrac{2^M+1}{2}$ | 1 | $2^M+1$ |
| Average Power | $\dfrac{1}{2}P_s$ | $\dfrac{2}{2^M+3}P_s$ | $\dfrac{1}{2^M}P_s$ | $\dfrac{2}{2^M+1}P_s$ | $\dfrac{2^M+1}{2^{M+1}}P_s$ | $\dfrac{1}{2}P_s$ |
| Transmission Bandwidth | $R_b$ | $\dfrac{2^M+3}{2M}R_b$ | $\dfrac{2^M}{M}R_b$ | $\dfrac{2^M+1}{2M}R_b$ | $MR_b$ | $\dfrac{2^M+1}{M}R_b$ |
| Channel Capacity | $C_b$ | $\dfrac{2M}{2^M+3}C_b$ | $\dfrac{2M}{2^M}C_b$ | $\dfrac{2M}{2^M+1}C_b$ | $MC_b$ | $\dfrac{M}{2^M+1}C_b$ |

When M = 1, the PWM mode requires a lower average power than other modulation methods, although the transmission bandwidth of the PWM mode is large and the channel capacity is low, while the system studied in this paper is mainly used to verify whether the proposed relay scheme is feasible, and initially realize the basic free-space optical communication, so that the transmission bandwidth and channel capacity of PWM mode meet the communication requirements. Compared with the other five modulation methods mentioned, the PWM method only needs to perform slot synchronization, without frame synchronization (symbol synchronization)—that is, each datum to be demodulated does not need to wait for timing, and only needs to wait for the rising edge of the detection signal, meaning that the PWM method will greatly reduce the modulation and demodulation difficulty of the communication system, but also reduce the error probability of data transmission, and achieve more reliable and stable communication compared with other modulation methods that need frame synchronization. The system implemented in this paper uses a repeater to complete point-to-multipoint free-space optical communication, and its implementation requires the ADC in the repeater to sample the received signal in all directions in a serial manner; if the pulse signal is narrow, a certain direction may not be able to sample the received signal, and then the wider pulse signal can increase the stability of the system. From Figure 1, it can be seen that under the condition of fixed time slot width, the PWM mode pulse width can be wider, compared with other modulation methods. PWM mode can make it easier for repeaters to sample the received signal, so PWM is more suitable for the research on networking technology of spatial optical communication than other modulation methods.

Figure 3 shows the PWM modulation and demodulation schematic, which is implemented through an FPGA. First, the binary data are discriminated; for datum 1, a pulse signal with a pulse width of T1 is generated by the FPGA, and for datum 0, a pulse signal with a pulse width of T2 is generated by the FPGA, and the pulse signal period is fixed to T. The demodulation process is implemented by the ADC, which has a sampling interval of τ in each direction during the fixed pulse signal period T time, and if there are N directions that need to probe the signal, the sampling rate of the ADC in the repeater is Nτ. In Figure 3, H represents when the sampled level signal is logic level 1, L represents when

the sampled level signal is logic level 0, and the number of instances of H is counted in T time. N1 represents the number of times the ADC samples to logic level 1 when the pulse width is $T_1$ in a pulse period T, and N2 shows the number of times the ADC samples to logic level 1 when the pulse width is $T_2$ in a pulse period T, and the data can be judged according to the size of the Count value.

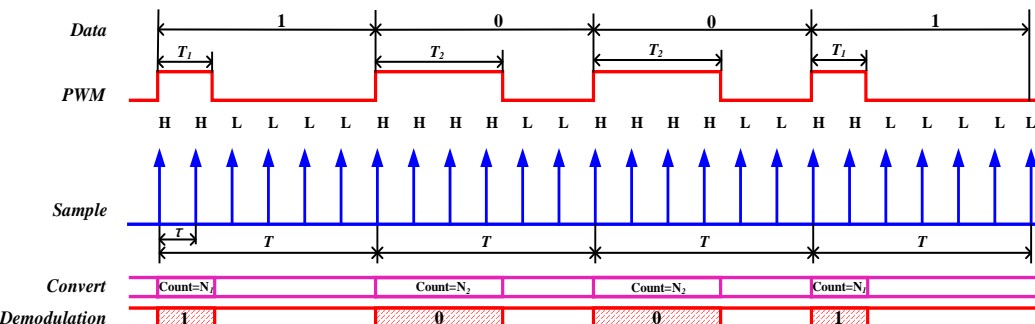

**Figure 3.** principle of pulse width modulation and demodulation [34].

This paper focuses on the design of a point-to-multipoint free-space optical communication system, which aims to provide an effective solution for free-space optical communication networking technology, and the experiment is carried out in a relatively ideal environment, so this paper does not consider the influence of atmospheric turbulence on the performance of a free-space optical communication system and the suppression of the atmospheric turbulence effect by adaptive optical technology. Assuming that only Gaussian white noise is added to the communication system, and the error caused by other factors is not considered, under the condition that the matching filter is an ideal linear filter, the input signal of the matching filter can be expressed as:

$$x(t) = \sqrt{P} + n(t) \tag{1}$$

where $P$ represents the peak optical power of the emitted light pulse, while $n(t)$ represents white Gaussian noise with a mean of 0 and a variance of $\sigma_n^2$. The resulting signal-to-noise ratio is:

$$SNR = \frac{P}{\sigma_n^2} \tag{2}$$

The probability of misidentifying datum 1 as datum 0 is set to $P_{1-0}$, and the probability of misidentifying datum 0 as datum 1 is set to $P_{0-1}$; then, we can obtain that [22,27]:

$$P_{1-0} = \frac{1}{2}[1 + erf(\frac{k - \sqrt{P}}{\sqrt{2\sigma_n^2}})] = \frac{1}{2}[1 + erf(\frac{k}{\sqrt{2\sigma_n^2}} - \sqrt{\frac{SNR}{2}})] \tag{3}$$

$$P_{0-1} = \frac{1}{2}[1 - erf(\frac{k}{\sqrt{2\sigma_n^2}})] \tag{4}$$

where the error function $erf(x) = \frac{2}{\sqrt{\pi}}\int_0^x e^{-t^2} dt$, $k$ is the decision threshold, and then the bit error rate of the PWM method is [44,47]:

$$P_{se} = \frac{1}{Z}[P_{1-0} + (Z-1)P_{0-1}] \tag{5}$$

where $Z = 2^M + 1$, therefore obtaining:

$$BER_{PWM} = \frac{1}{2^{M+1}}[1 + erf(\frac{k}{\sqrt{2\sigma_n^2}} - \sqrt{\frac{SNR}{2}}) + (2^{M+1} - 1)erf(\frac{k}{\sqrt{2\sigma_n^2}})] \tag{6}$$

According to $Q = \sqrt{2}\mathrm{erfcinv}(2BER)$, it further is obtained that:

$$Q = \sqrt{2}\mathrm{erfcinv}(\frac{1}{2^M}[1 + erf(\frac{k}{\sqrt{2\sigma_n^2}} - \sqrt{\frac{SNR}{2}}) + (2^{M+1} - 1)erf(\frac{k}{\sqrt{2\sigma_n^2}})]) \tag{7}$$

B.    Theory of laser relay station networking

Figure 4 is a schematic diagram of the structure of the laser relay station networking system we constructed. The system consists mainly of a repeater and multiple transceivers. The repeater can simultaneously realize the integration of the transceiver in any direction on the same receiving level, and the transceiver can transmit data in any direction within 360° centered on the repeater, while the repeater can send data to other transceivers in any direction, without considering the high-precision alignment between the transmitter end and the final receiving end, and also avoid the signal interference problem when multiple devices in different positions are cross-laser communication.

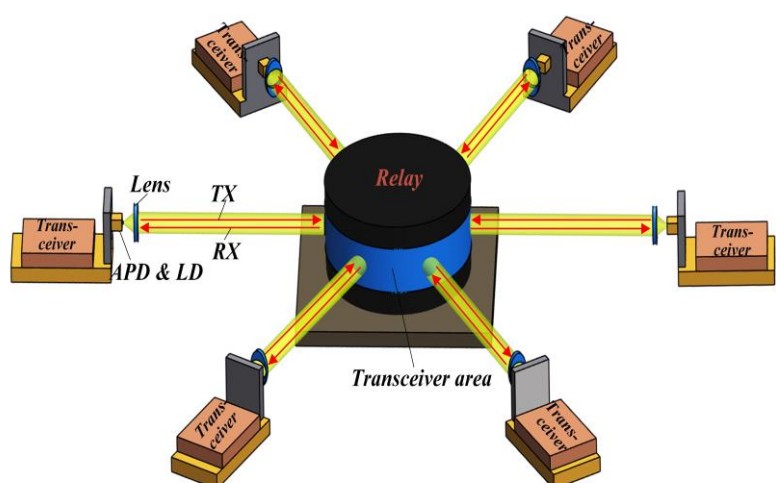

**Figure 4.** Schematic diagram of the structure of the laser relay station networking system.

The transceiver can be placed anywhere around the repeater, the repeater is responsible for transmitting data, and the transceiver can send data to the repeater or receive the data forwarded by the repeater. When the transceiver transmits data, it first encodes the address of the data to be transmitted, specifies the position of the receiving end, and then modulates the encoded data. The data are sent in the form of a laser signal through laser secondary light (LD), realizing the conversion from a digital signal to an optical signal. In Figure 4, TX represents the laser signal sent by the transceiver to the repeater, which contains address information and data. The laser transceiver area in the repeater contains an avalanche photodiode and laser diode, which can be used for laser transmission and laser reception, realizing the function of the integrated transceivers. In the repeater, it is neces-

sary to scan and control the photodetectors in all directions, so as to receive the laser signals in different directions. The repeater demodulates the received laser signal, parses the address information and data information, sends the data to the transceivers in different directions according to the address information, realizes the function of address routing, and also completes the conversion of the laser signal into a digital signal. Therefore, the data demodulation and re-modulation are completed in the repeater. TX represents the laser signal sent by the repeater to the transceiver. This signal is a re-modulated optical signal. Because they specify the specific direction of the receiver, the RX signals only contain data. When the transceiver is used as a receiver, it is only responsible for receiving the laser signal sent by the repeater and demodulating it. The demodulated data are the data that the transmitter wants to send to the target receiver transmitter, enabling laser communication between the transceivers in different directions.

Figure 5 shows the internal structure of the laser relay station. The data routing module is responsible for selecting the data path and specifying the receiving and transmitting paths. The data forwarding module completes the data demodulation and re modulation, realizes the forwarding of the received data, and selects the forwarding path according to the data routing module. The laser transceiver area is composed of a lens array, an avalanche photodiode and a laser diode. The avalanche photodiode and laser diode are close to each other to achieve laser reception and transmission through an optical lens. If the divergence angle of the laser diode selected is 30°, and the photosensitive area of the avalanche photodiode is 500 μm, the placement angle of the lens array can be shown in Figure 5. Each lens in the lens array is closely arranged, and 360° data can be transmitted and received to further complete communication with transceivers in any direction.

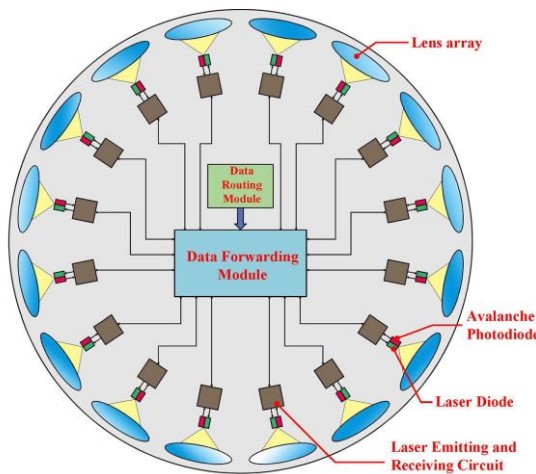

**Figure 5.** Internal structure diagram of the laser relay station.

Figure 6 shows the data address routing scheme. The input signal in any direction can be output in any direction after the address routing selection, and the specific configuration is selected according to the transceiver address, or it can be set during the human–computer interaction of the repeater; here, the address routing scheme in six directions is given. The address routing process is completed through a switch array—each switch has a different state, the input signal can choose a different path for output, and the specific implementation process is implemented in the FPGA. Each switch is composed of two two-choice data selectors; when the control terminal Select is high, the output end out_a corresponds to the input end in_a, and the output end out_b corresponds to the input end in_b; when the control terminal Select is low, the output end out_a corresponds to the input end in_b, and the output end out_b corresponds to the input end in_a, meaning that each switch has two working states. Each switch in the switch array with different states is combined with each other, and the signal output of any path can be realized. The output end in each direction corresponds to an address number. The signal input end in each

direction will carry a piece of address information. The address router will configure the switch array according to the address information, so as to output the input signals in different directions along the specified direction. If the sent data do not carry address information, we can configure the input and output paths in human–computer interaction, and manually select to output the input signal in the specified direction.

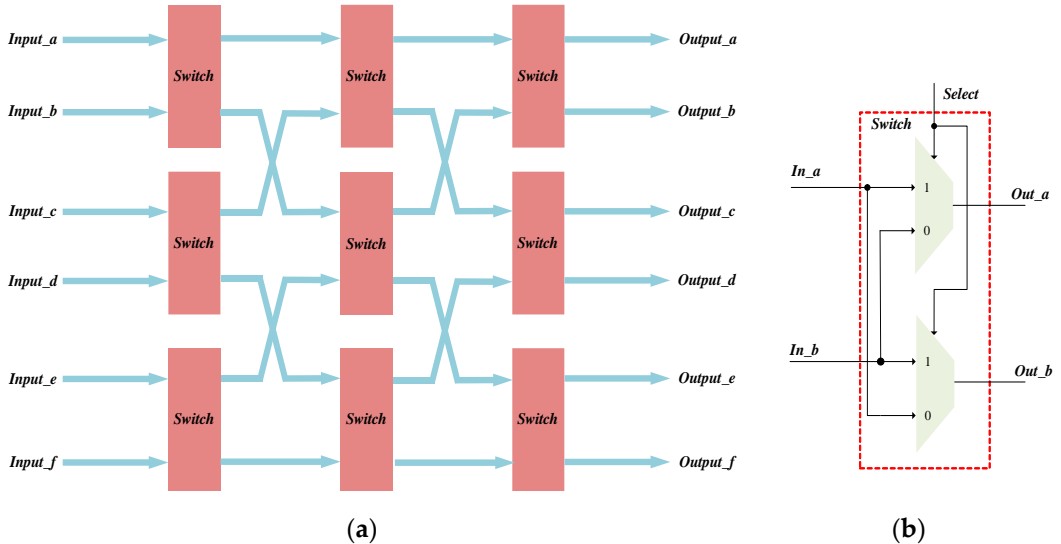

**Figure 6.** Principle of address routing. (**a**) Address routing structure. (**b**) Switch structure.

C. Design of system

Figure 7 is a block diagram of the hardware composition of the point-to-multipoint free-space optical communication system, which aims to transmit the data from the transmitter to the receiver in different directions through the repeater. In the laser emission system, the data in the microprocessor (MCU) are first sent to the data transmission module of FPGA, and the data transmission module sends the obtained data to the PWM modulator for modulation. The modulated signal drives the laser emission circuit. The optical communication relay system can receive laser signals in all directions. The relay controller realizes the functions of ADC sampling, data demodulation, address distribution, data remodulation, and the control of the receiving and transmitting directions of the received signals in all directions. The modulated data drive the laser transmitting circuits in different directions according to the distributed addresses. The receiving end detects the laser signal through APD, the laser signal is sent to the PWM demodulator for demodulation through the receiving circuit, and the demodulated data are sent to MCU through the data receiver, thus completing point-to-multipoint free-space optical communication.

A diagram of the operation of the relay station is shown in Figure 8, mainly composed of a direction selector and a relay controller (inside FPGA). The direction selector is mainly composed of ADC and analog switches. RX represents the sampling signal of ADC, which controls the ADC to sample the received signals (RX_1~RX_n) in different directions by controlling the analog switch. TX represents the driving signal of the laser transmitting circuit, which drives the LD in different directions by controlling the analog switch driving signals (TX_1~TX_n). The relay system requires fast ADC sampling. Here, AD9226 from Analog Devices (ADI) Semiconductor Company was selected to achieve a sampling frequency of 65M. The relay controller is mainly composed of a PWM demodulator, an address resolver, a PWM modulator, and a direction controller. The principles of the PWM modulator and the PWM demodulator are shown in Figure 9 and Figure 10, respectively. The direction controller controls the analog switch to further control the sampling direction and laser emission direction of ADC. The sampling results (data) of ADC are

directly sent to the PWM demodulator for data discrimination and further data demodulation. Point-to-multipoint communication involves each group of data having a specific code; the demodulated data are sent to the address parser, the transmission direction target is selected according to the different coding addresses of the data, and the data are sent to different receiving targets through the PWM modulator, thus realizing point-to-multipoint free-space optical communication.

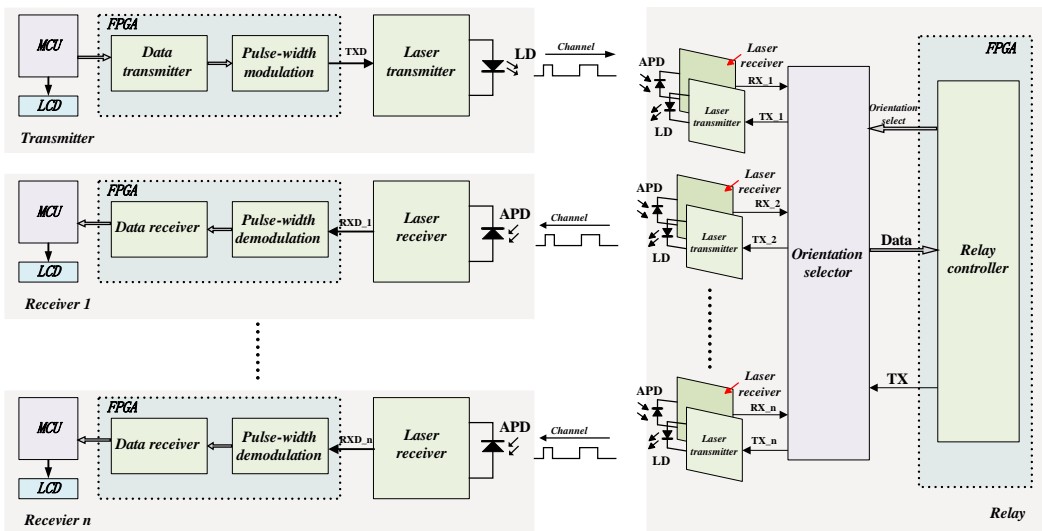

**Figure 7.** Hardware composition block diagram of a point-to-multipoint free-space optical communication system.

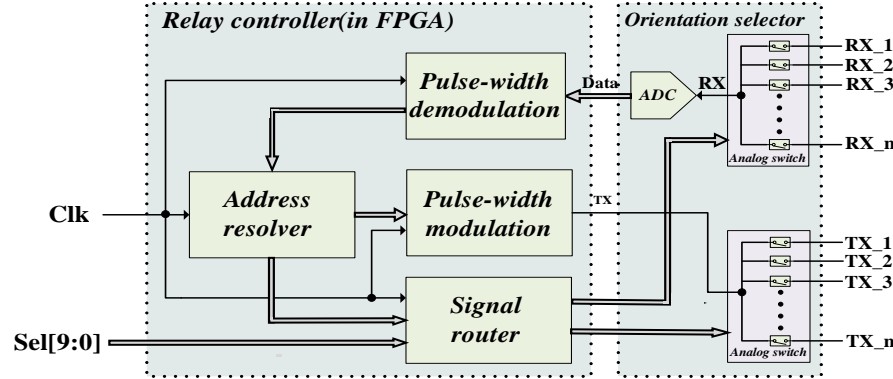

**Figure 8.** Diagram of the operation of the relay station.

Figure 9 is the system block diagram of the transmitting device, which is mainly composed of a PWM modulator and a laser transmitting circuit. The PWM modulator is contained inside the FPGA, which is mainly composed of a PWM pulse generator, a data converter, and an optional data selector. The laser emission circuit is composed of a voltage-controlled current source, N-MOSFET, and a laser diode (LD). The data input to the PWM modulator is data with a bit width of 8. The data converter converts the 8-bit parallel data into serial data and modulates the converted data bit by bit. If the data belong to datum 1, the PWM pulse generator will generate a narrower pulse signal and send it through the data selector. If they belong to datum 0, the PWM pulse generator will generate a wider pulse signal and send it through the data selector, and the PWM signal TX generated by the PWM modulator drives N-MOSFET. When TX is logic level 1, N-MOSFET is on, the voltage-controlled current source generates a constant current, and the LD emits light. When TX is logic level 0, N-MOSFET is off, and the LD does not emit light. Therefore, the transmitting system realizes the function of modulating data into a PWM laser signal.

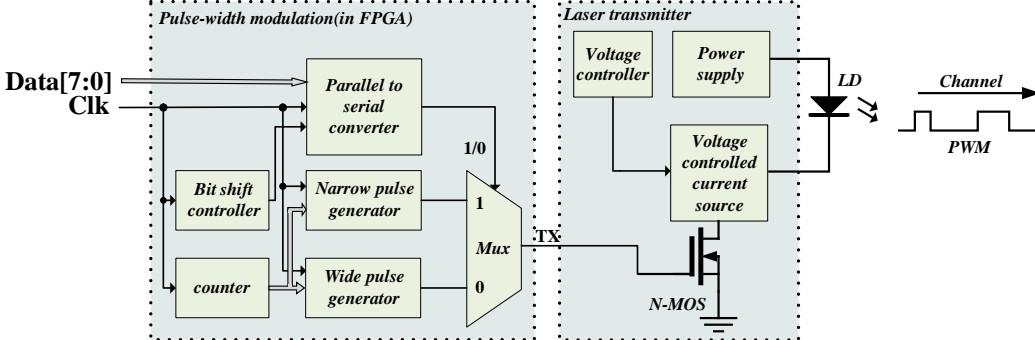

**Figure 9.** Pulse width modulation and transmitting circuit.

The circuit principle of the receiving device is shown in Figure 10, which is mainly composed of a laser receiving circuit and a PWM demodulator (inside the FPGA). The laser receiving circuit is mainly composed of an avalanche photodiode (APD), a trans-impedance amplifier (TIA), and a reverse proportional operational amplifier. The PWM demodulator is mainly composed of an ADC sampler, logic discriminator, pulse width discriminator, shift controller, counter, data discriminator, and data converter. The APD photocurrent I received by the pulse laser receiving circuit is converted into voltage $V_1$, and then the voltage $V_1$ is further amplified by the second amplifier to obtain voltage $V_2$, so the laser signal is converted into the voltage signal RX, where the voltage $V_2$ is:

$$V_2 = -\frac{R_3}{R_2}V_1 = -\frac{R_3}{R_2}\left(-IR_1\right) = \frac{R_1 R_3}{R_2}I \tag{8}$$

ADC samples the RX signal, converts the voltage signal into data and sends the sampled data to the PWM demodulator.

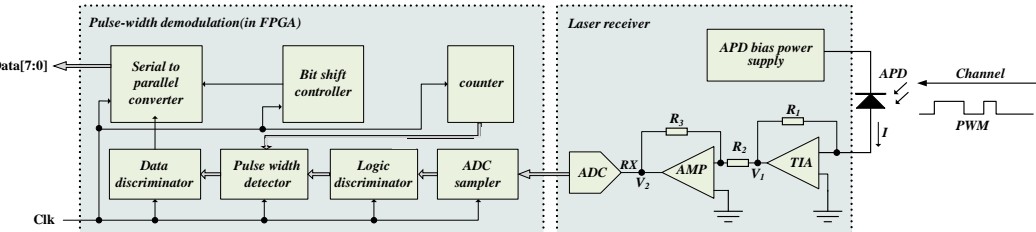

**Figure 10.** Pulse width demodulation and receiving circuit.

In the PWM demodulator, the ADC sampler drives ADC to sample the RX signal. The logic discriminator judges the logic level according to the data obtained. When logic level 1 is detected, the counter starts counting. When logic level 0 is detected, the counter stops counting. At the same time, the pulse width detector counts the logic judgment results, the data discriminator judges the data according to the statistical results of the pulse width detector, and finally converts the serial 8-bit data to the parallel 8-bit data output through the shift controller and data converter.

## 3. Experiment and Analysis

Figure 11 shows the test diagram of the point-to-multipoint free-space optical communication system. We mainly verified that the data at the transmitting end were sent to the receiving end in different directions through the repeater. Considering the production cost and the convenience of the test, we created a simple optical communication system that can send and receive data in six directions. We completed the basic prototype of the optical communication network group technology and verified it. The system mainly included a laser transmitter, a communication relay and three laser receivers. The hardware circuit mainly included a laser transmitting circuit, a laser receiving circuit and a system

control circuit. A continuous laser diode was selected as the emission light source in the laser emission circuit. Considering the safety of human eyes during the experiment, the emission power of the laser diode was 200 mW, and the wavelength was 940 nm. The voltage-controlled current source chip IC-HK of iC Haus Company was used to drive the laser diode. The maximum switching frequency of 155 MH was achieved to meet the communication requirements. LSSAPD9-500 was selected as the photodetector of the laser receiving circuit. The APD had high reliability and low dark current, with a response band of 400–1100 nm and a high response to 940 nm laser, which could improve the reliability of the communication system. In the relay system, LD and APD shared a focus lens. They fitted together and were in the focus position of the lens, saving hardware resources and realizing the function of integrated transceiver. The system control circuit sampled FPGA and MCU as the main control chip. FPGA was responsible for data modulation, data demodulation, data address allocation, etc. MCU was responsible for communication debugging, displaying the sent data or received data on the liquid crystal display (LCD). The system displayed the data on the LCD in the form of two-dimensional images. The FPGA was EP4CE55F23C6N from Altera Company, and the MCU was STM32F103ZET6 from Italy France Semiconductor Company.

In the system, the test diagram of which is shown in Figure 11, we sent three 250 × 250-pixel color pictures with 16-bit color. The picture contents were pens, pavilions, and cartoon characters. Without moving the transmitter transmission angle, we needed to send the picture with pen contents to receiver A, the picture with pavilion contents to receiver B, and the picture with cartoon characters to receiver C. The three receivers were in different directions. We fixed the repeater on the optical platform. The transmitter was 25 cm away from the repeater, receiver A was 27 cm away from the repeater, receiver B was 30 cm away from the repeater, and receiver C was 26 cm away from the repeater. The communication distance was limited by the transmission power of the LD. Increasing the transmission function power increased the communication distance. Considering the safety of the experiment, we chose an LD with smaller transmission power and a shorter communication distance to complete the experiment. In order to verify the point-to-multipoint optical communication system, we sent three pictures to three different receiving targets through the relay system. The experimental results show that the picture content received by receiver A was a pen, the picture content received by receiver B was a pavilion, and the picture content received by receiver C was a cartoon character. Therefore, the system realized point-to-multipoint optical communication. At the same time, compared with the point-to-point optical communication system, it was more flexible and efficient.

In order to verify the communication quality, we used an oscilloscope to detect the signal waveforms of the transmitting end and the three receiving ends, respectively. The three pictures were sent out serially in pixel units. One pixel in the picture was composed of 16-bit data. Each pixel was divided into two groups of 8-bit data and sent out successively. Each group of data has a fixed code to represent the address of the receiver. The repeater retransmits the data to different receiving targets according to different coded information. In Figure 12, the yellow signal represents the signal waveform at the transmitting end, the blue signal represents the signal waveform received by receiver A, the red signal represents the signal waveform received by receiver B, and the green signal represents the signal waveform received by receiver C. The modulation and demodulation of the signal are realized by a finite state machine. The data bit width transmitted by the transmitting end each time is 10 bits, where the first two bits represent the receiver address code, and the last eight bits represent the data of the picture. The repeater demodulates the signal at the transmitting end, selects the address routing, and automatically extracts the picture data and sends them to the designated receiving target. The receiving data at the receiving end are 8 bits in size. The address code of receiver A is 01, that of receiver B is 10, and that of receiver C is 11. 01_1101_1110 represents the data sent by the transmitter in Figure 12, in which 01 represents the address information of receiver A. After the relay system recognizes the address information, the data 1101_1110 are sent to

receiver A. At this time, the data received by receiver A do not contain address information.

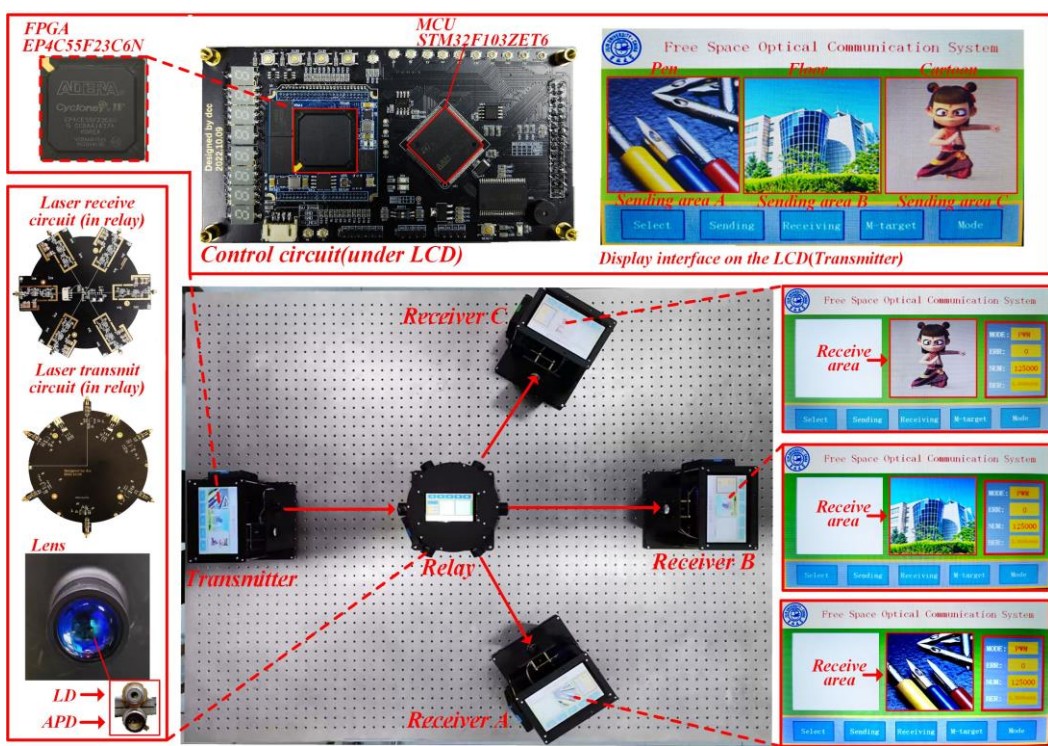

**Figure 11.** Test diagram of the free-space optical communication system.

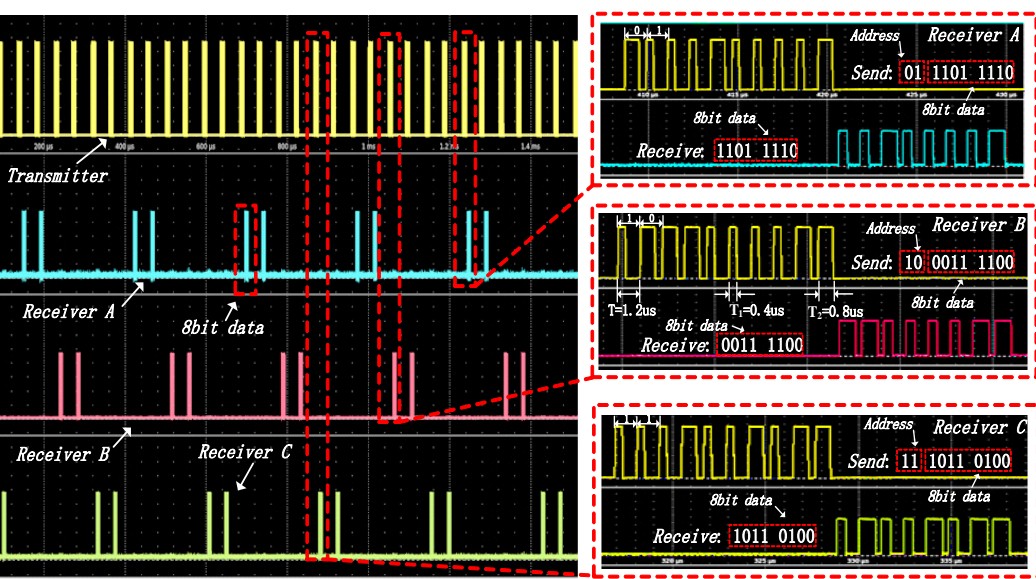

**Figure 12.** Test waveform of free-space optical communication system with point-to-multipoint.

In the experiment, the cycle of PWM was T = 1.2 μs, the pulse width time of characterization datum 1 was $T_1$ = 0.4 μs, and the pulse width time of characterization datum 0 was $T_2$ = 0.8 μs. Therefore, the transmitting end needed at least 1.2 μs to send 1-bit data, and the receiving end needed at least 0.6 μs to demodulate 1-bit data. Since the relay system needed to receive complete 10-bit data and send them to the receiving target through address judgment, the signal time difference between the transmitting end and the final receiving end was 11.975 μs. Since the transmission power of the light source is 200 mW, the average output power of the PWM signal is:

$$P_{\text{PWM}} = (\frac{T_1}{T} + \frac{T_2}{T})P_s = 200\text{mW} \tag{9}$$

It takes 10 cycles to send 1 byte of data, and so the data transmission bandwidth is:

$$BW_{\text{PWM}} = \frac{1}{10T} \approx 81.38\text{KB/s} \tag{10}$$

Reducing the pulse width of PWM can reduce the average output power. For example, $T_1 = 0.2\ \mu s$, $T_2 = 0.4\ \mu s$, then the average transmission power becomes 100 mW. However, during modulation, we needed to make the characteristics of datum 1 and datum 0 more distinctive—that is, make the pulse width difference between the narrow pulse and the wide pulse more obvious. Since the communication is performed in bad conditions, the too small pulse width signal will be demodulated after APD detection, which will increase the error probability, Therefore, we sacrificed the average transmission power and increased the pulse width to ensure the stability of the communication system.

To verify the stability of the system, we gently moved the three receivers so that the transmission light source of the relay system was not directly aimed at the lens of the receiver. In this way, the pulse signal received by the receiver was not a signal with high energy, so the amplitude of the pulse received by the receiver could be controlled. In this way, we simulated poor communication conditions. In Figure 13, we control the amplitude of the received pulse voltage of receiver A to be about 2.055 V, that of receiver B to be about 1.527 V, and that of receiver C to be about 1.406 V. The yellow pulse signal represents the drive signal of LD at the transmitting end, with a pulse amplitude of 3.3 V. The blue signal represents the signal received by receiver A, the red signal represents the signal received by receiver B, and the green signal represents the signal received by receiver C. The 13-bit parallel ADC was used in this system. The highest bit represents the symbol bit, and the sampling voltage range is −5 V~+5 V. After the ADC was calibrated, we set the threshold voltage of the system discrimination data to about 1.17 V.

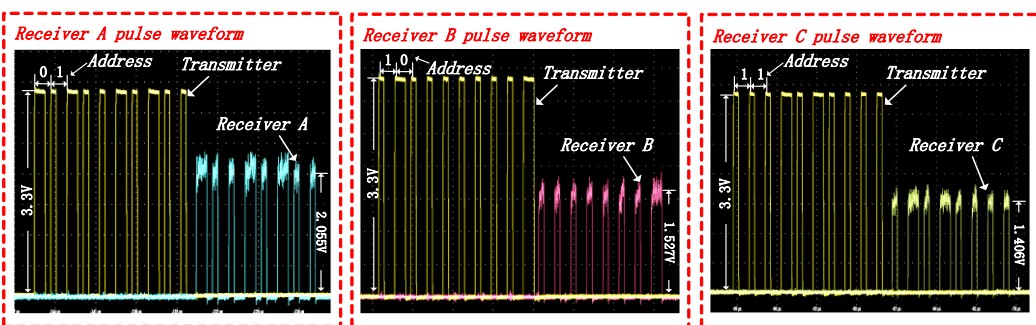

**Figure 13.** Waveform diagram of receiving terminal for changing receiving angle.

When the signal waveform at the receiving end was not ideal, we demodulated the received signal with ADC (ADC samples the signal and sends it to FPGA) and the signal directly with FPGA. When the signal was demodulated directly with FPGA, we directly sent the received pulse signal to FPGA to detect its edge and then used a counter to measure the pulse width to demodulate the data. Figure 14 shows the picture transmission and bit error rate of three receivers under three conditions. The ADC was used for demodulation, and the three receivers received the corresponding pictures accurately without error. When the FPGA was used to directly demodulate the signal when the voltage at the receiving end was 2.055 V, receiver A successfully received the picture and there was no bit error. When the voltage at the receiving end was 1.527 V, receiver B successfully received the picture, but the bit error rate was high, at about 0.37437600. There were many white dots in the picture, which is different from the original picture. When the voltage at the receiving end was 1.406 V, the bit error rate of receiver C was 0.99958403, and the picture

could not be received, the receiving area was blank. At this time, the FPGA could not recognize the pulse signal, resulting in the failure of information transmission.

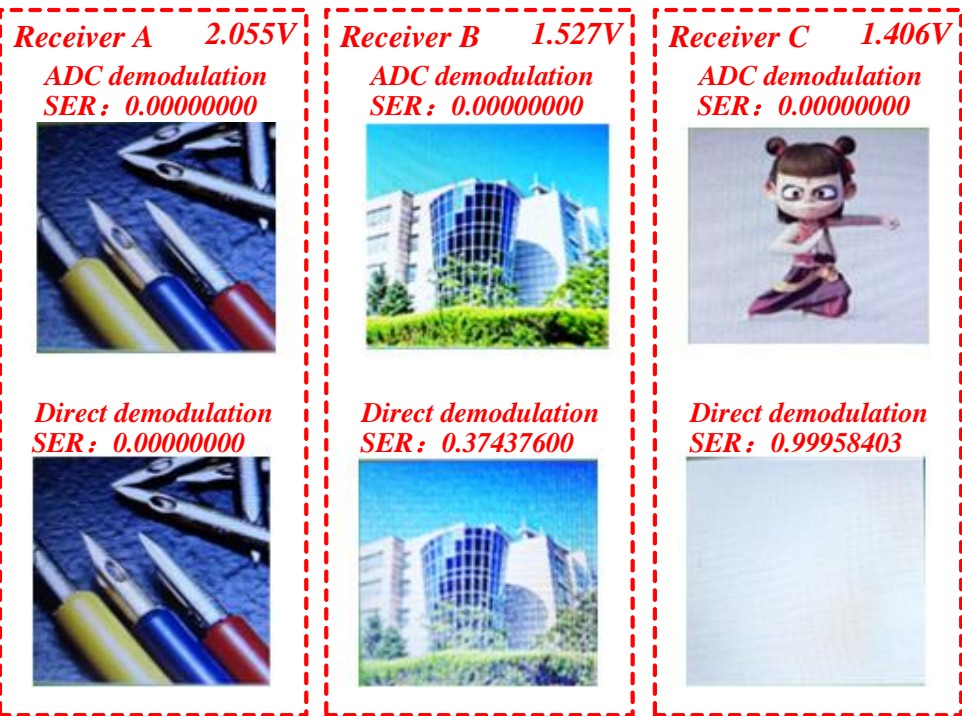

**Figure 14.** Picture transmission test chart for changing reception angle.

Since the hardware circuit, mechanical structure and electronic components of the three receivers were almost identical, in order to further verify the impact of ADC demodulation on system stability, we conducted many experiments with receiver A. It can be seen from the experimental results in Table 2 that when the pulse amplitude voltage at the receiving end is between 1.661 V and 2.055 V (under the condition of a good communication environment), ADC sampling for demodulation and FPGA direct demodulation can accurately transmit data without bit error rate; When the pulse amplitude voltage at the receiving end gradually decreases from 1.585 V (in the case of poor communication environment), the bit error rate of the FPGA direct demodulation mode begins to appear, and with the gradual decrease in the pulse amplitude voltage at the receiving end, the bit error rate gradually increases, and when the pulse amplitude voltage at the receiving end decreases to 1.413 V (under the condition of a harsh communication environment), the bit error rate of the FPGA direct demodulation mode reaches 100%; However, for the method of demodulation using ADC sampling, when the pulse amplitude voltage at the receiving end is reduced to 1.413 V, the system has a lower bit error rate, and when the pulse amplitude voltage at the receiving end is reduced to 1.007 V, the system has a bit error rate of 100%. Through experimental data, we can analyze that in the case of a good communication environment, the two demodulation methods are no different for this system; in the case of a poor communication environment, the ADC sampling method for demodulation can accurately transmit data. At this time, the FPGA sampling mode has a higher bit error rate, and the data transmission is inaccurate. In the case of a harsh communication environment, although the bit error rate occurs in the demodulation method using ADC sampling, the data can still be transmitted, and the FPGA direct demodulation method cannot transmit data. Therefore, this experiment proves that demodulation using ADC sampling can increase the stability of the communication system, and further proves that the scheme proposed by us using pulse width modulation technology to implement a point-to-multipoint free-space optical communication system is feasible.

**Table 2.** Bit error rate corresponding to different pulse amplitudes at the receiver.

| Method of demodulation | Bit Error Rate (%) | | | | | | | |
|---|---|---|---|---|---|---|---|---|
| | 2.055 V | 1.914 V | 1.859 V | 1.716 V | 1.661 V | 1.585 V | 1.544 V | 1.508 V |
| ADC demodulation | 0 | 0 | 0 | 0 | 0 | 0 | 0 | 0 |
| Direct demodulation | 0 | 0 | 0 | 0 | 0 | 2.75920 | 3.39679 | 98.95840 |
| Method of demodulation | Bit Error Rate (%) | | | | | | | |
| | 1.413 V | 1.364 V | 1.311 V | 1.287 V | 1.233 V | 1.197 V | 1.007 V | 0.965 V |
| ADC demodulation | 0.001605 | 16.64621 | 29.18241 | 38.39984 | 44.65121 | 99.53920 | 100 | 100 |
| Direct demodulation | 100 | 100 | 100 | 100 | 100 | 100 | 100 | 100 |

## 4. Discussion

Table 3 compares some of the performances of the free-space optical communication system, mainly including the response time of the receiving end, whether frame synchronization is required, and whether mechanical alignment is required. It is obvious that our system does not need frame synchronization when realizing point-to-multipoint free-space optical communication, greatly simplifying the demodulation circuit, and does not need to perform mechanical scanning to lock the direction of data transmission, demonstrating certain advantages compared with other free-space optical communication systems. In addition, compared with point-to-multipoint free-space optical communication systems that require mechanical alignment, our system has obvious advantages in its response time at the receiving end. Although the point-to-point free-space optical communication system has a fast response speed, its function is singular and it cannot complete point-to-multipoint data transmission, so our system has certain advantages in realizing point-to-multipoint free-space optical communication.

**Table 3.** Performance comparisons of the reported laser communication system.

| Structures of FSO Communication System | Multi-Destination Data Transfer | Response Time | Frame Synchronization | Mechanical Alignment | Refs |
|---|---|---|---|---|---|
| Rotating double prism | Yes | 1.5 s | Yes | Yes | [3] |
| magnetometer sensors | Yes | <5 s | — | Yes | [7] |
| VIPA-based 2D optical beam-steering technique | Yes | — | Yes | No | [11] |
| Point-to-Point System | No | 28.87 ns | No | No | [34] |
| wavelength-division multiplexing | Yes | — | Yes | No | [46] |
| Relay system | Yes | 11.957 us | No | No | Our work |

## 5. Conclusions

In this paper, we use PWM modulation technology combined with the data routing and forwarding capabilities of laser relay stations to realize free-space optical communication networking, and use high-speed ADC to demodulate the received signal, which increases the stability of the free-space optical communication system. A series of two-dimensional image transmission experiments and bit error rate test experiments are carried out on the system. The experimental results show that the system is more flexible than the point-to-point laser communication system, and the communication quality is higher. At the same time, the system realizes point-to-multipoint free-space optical communication without adjusting the transmission angle, we believe that the system proposed in this paper can provide an effective solution for free-space optical communication networking technology.

**Author Contributions:** methodology, C.D.; software, C.D.; validation, C.L.; formal analysis, Z.W.; investigation, Z.G.; resources, Z.L.; data curation, J.S.; writing—original draft preparation, C.D.; writing—review and editing, M.T.; supervision, J.S.; project administration, M.T.; funding acquisition, J.S. All authors have read and agreed to the published version of the manuscript.

**Funding:** This research was funded by the National Natural Science Foundation of China under Grants No. 62090053, No. 62090054 and No. 61934003; The Major Scientific and Technological Program of Jilin Province under Grant No. 20200501007GX; The Project of Industrial Technology Research and Development of Jilin Provincial Development and Reform Commission under Grant No. 2020C019-2 and the Program for JLU Science and Technology Innovative Research Team under Grant No. 2021TD-39.

**Institutional Review Board Statement:** Not applicable.

**Informed Consent Statement:** Not applicable.

**Data Availability Statement:** The raw/processed data required to reproduce these findings cannot be shared at this time as the data also forms part of an ongoing study.

**Acknowledgments:** We are grateful to the anonymous reviewers for constructive comments that improved the quality of the work.

**Conflicts of Interest:** The authors declare no conflict of interest.

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
