# Peer review of "Free Space Optical Communication Networking Technology Based on a Laser Relay Station"

_applsci, doi:10.3390/app13042567_

Round 1

Reviewer 1 Report

The authors demonstrated free space optical communication networking using a laser relay station. They Although the idea is good but the system has been implemented for short-range free space optical communication. As a result, the proposed model may not be suitable for commercial applications. There are many typos/grammatical mistakes in the manuscript. I would recommend the authors to address the following suggestions in the revised manuscript.

Abstract:

(1) ……is used to realize space optical communication….. à is used to realize free space optical communication

(2) …. can achieve free optical communication between various…. à can achieve free space optical communication between various

Introduction:

…………. In order to realize the space optical communication……… à In order to realize the free space optical communication

Theory and system design:

(1) Figure 1/Figure 2/Figure 3: Citation is missing in the figure captions.

(2) The position of the transceiver can be placed at will, and each transceiver can conduct two-way data transmission, which can not only send data information but also receive data information. à The sentence is not clear

(3) …… the point-to-multipoint space optical communication system….. à the point-to-multipoint free space optical communication system

Experiment and analysis:

(1) ….. diagram of the space optical communication system….. à diagram of the free space optical communication system

(2) The authors used a laser diode having operating wavelength of 940 nm which is unconventional. They should consider one of the three optical windows. I also recommend the authors to follow ITU-T/IEEE standard for designing and implementing the proposed model.

(3) The distance between the transmitter, relay, and, receiver is too short. The authors should focus on long distance free space optical communication.

(4) The transmission bandwidth of the proposed device is too low to be considered for commercial application.  

(5) The information regarding Bit Error Rate, Signal-to-noise Ratio, and Q-factor are missing in the manuscript.

(6) The authors are encouraged to check the eye diagram after the signals are received in the photodetector. 

(7) The authors are encouraged to provide a comparison table to compare their results with other reported research works.

Reviewer 2 Report

Зазук is interesting and I hope should be very useful for the scientists and end-users how would like to do research in this field. I really liked very much the introduction, the first part of the paper. But I have one, I think important comment - authors in some details explained the advantages as well as problems of the use of free space communication systems but they did not mention the influence of atmospheric disturbances (like turbulence or scattering) on the efficiency of the signal transportation. But this greatly reduce distance of the use of any communication system. And the only way to fight or correct the influence for sure in to use adaptive optics - in particular high-speed adaptive optical systems. Such systems, working with the speed of up to 2 kHz are now developed in Australia as well as ... in Russia. There are some papers in Optics Express about the use of adaptive optics to correct for laser beam disturbances caused by scattering process. I think this really should be mentioned in the preface. 

Reviewer 3 Report

1. Which modulation technique is good? why?

2. In Fig 5, how the array angle is defined?

3. Write the inference of table 2 and highlight the better outcomes of the table.

4. Add latest article in the literature survey

5.  Rewrite the abstract and conclusion in concise form

Round 2

Reviewer 1 Report

Not Applicable.